# Interstitial lung disease diagnosis and prognosis using an AI system integrating longitudinal data

Xueyan Mei [1] ✉, Zelong Liu [1], Ayushi Singh[2], Marcia Lange [3], Priyanka Boddu[3], Jingqi Q. X. Gong[4], Justine Lee[2], Cody DeMarco[2], Chendi Cao[1], Samantha Platt [3], Ganesh Sivakumar[3], Benjamin Gross[3], Mingqian Huang[2], Joy Masseaux[2], Sakshi Dua[5], Adam Bernheim[2], Michael Chung [2], Timothy Deyer[6,7], Adam Jacobi[2], Maria Padilla[5], Zahi A. Fayad [1,2] ✉ & Yang Yang [1,2,8] ✉

For accurate diagnosis of interstitial lung disease (ILD), a consensus of radiologic, pathological, and clinical findings is vital. Management of ILD also requires thorough follow-up with computed tomography (CT) studies and lung function tests to assess disease progression, severity, and response to treatment. However, accurate classification of ILD subtypes can be challenging, especially for those not accustomed to reading chest CTs regularly. Dynamic models to predict patient survival rates based on longitudinal data are challenging to create due to disease complexity, variation, and irregular visit intervals. Here, we utilize RadImageNet pretrained models to diagnose five types of ILD with multimodal data and a transformer model to determine a patient's 3-year survival rate. When clinical history and associated CT scans are available, the proposed deep learning system can help clinicians diagnose and classify ILD patients and, importantly, dynamically predict disease progression and prognosis.

Interstitial lung disease (ILD) refers to a group of more than 200 pulmonary conditions which can exhibit varying degrees of lung parenchymal fibrosis[1]. Obtaining a specific diagnosis in cases of ILD is essential to guide patient management and treatment. High-resolution computed tomography (HRCT) plays a significant role in accurately classifying the various subtypes of ILD. According to the American Thoracic Society (ATS) guidelines, accurate diagnosis of ILD subtypes requires a multidisciplinary assessment reviewing clinical history, HRCT, and pathology[2]. In addition, longitudinal monitoring with CT can assess the progression of Serial CT that can reveal changes in the extent

of parenchymal architectural distortion, reticulation, bronchiectasis and honeycombing, allowing for the identification of progressive fibrotic disease which correlates with poorer survival. Mortality is often not feasible as an end-point for diseases with chronic progressive fibrosis (such as IPF); change or lack of change in disease extent on HRCT represents a potential means of assessing treatment response[3].

In some cases, diagnosis and classification of ILD types via CT are relatively straightforward. In other cases, the imaging findings can overlap multiple ILD patterns or may have no identifiable pattern, and is thus subject to substantial inter- and intra-observer variation among

[1]BioMedical Engineering and Imaging Institute, Icahn School of Medicine at Mount Sinai, New York, NY, USA. [2]Department of Diagnostic, Molecular, and Interventional Radiology, Icahn School of Medicine at Mount Sinai, New York, NY, USA. [3]Icahn School of Medicine at Mount Sinai, New York, NY, USA. [4]Department of Pharmaceutical Sciences, Icahn School of Medicine at Mount Sinai, New York, NY, USA. [5]Department of Medicine, Pulmonary, Critical Care and Sleep Medicine, Icahn School of Medicine at Mount Sinai, New York, NY, USA. [6]Department of Radiology, Cornell Medicine, New York, NY, USA. [7]Department of Radiology, East River Medical Imaging, New York, NY, USA. [8]Department of Radiology and Biomedical Imaging, University of California, San Francisco, CA, USA. ✉e-mail: xueyan.mei@icahn.mssm.edu; zahi.fayad@mssm.edu; yang.yang4@ucsf.edu

radiologists[4]. Interpretation of these difficult exams is challenging and can depend on the expertize of the radiologist.

Prior studies have shown that deep learning can be used to recognize different ILDs on CT images[5], including detecting abnormal interstitial patterns[6], automatic assessment of the extent of systemic sclerosis-related ILD[7], and differentiation between nonspecific interstitial pneumonia (NSIP) and usual interstitial pneumonia (UIP)[8]. However, literature regarding accurate deep learning-aided diagnosis of multiple ILD subtypes as well as prediction of survival rate is limited at this time. The purpose of our study is to develop an AI system that (1) can classify 5 different types of ILD based on initial chest CT scans and relevant clinical history as well as (2) monitor a patient's disease progression.

For our study, we collected clinical information retrospectively through a chart review of electronic medical records. Clinical information included age, sex, history of current/former smoking, history of rheumatic disease, home oxygen requirement, history of occupational exposures, pulmonary function test (PFT) values (FEV1/FVC ratio, FEV1 value, DLCO percentage), presence of pulmonary hypertension based on echocardiography or right heart catheterization, and history of lung biopsy. Clinical history was collected longitudinally for every CT scan available for the patient through the course of their treatment to account for changes in exposures or other variables. We collected CT scans and the corresponding clinical history obtained at every clinical encounter. To further predict a patient's 3-year survival rate, we included medications and other therapeutic information to clinical history (Fig. 1 and Fig. 2).

For subtype classification, we first created a deep convolutional neural network (CNN) and a vision Transformer[9] (ViT) to learn image patterns of patients with ILD on the initial chest CT scan. We then used multilayer perceptron[10] (MLP), XGBoost[11], and support vector machine[12] (SVM) classifiers to predict ILD subcategories based on clinical information. Finally, we developed a joint model integrating chest CT characteristics with associated clinical history to predict ILD subtypes. To predict a patient's survival within 3 years from the initial visit, we created Transformer[13] and long-short-term memory[14] (LSTM) models to study longitudinal CT scans and longitudinal clinical information. The joint CNN model and the Transformer models showed the best scores on the validation set. Hereafter, the performance of the joint CNN and Transformer models is reported. The performances of other models can be found in Supplementary Figs. 2–6.

## Results

The Mount Sinai Medical Center Research Registry for Interstitial Lung Disease (MSMC-ILD) was established in 2014. Patients enrolled in MSMC-ILD had a consensus diagnosis from radiology, pathology, and pulmonology. 449 patients with 1822 CT scans were collected between September 2014 and April 2021 from 230 centers in the United States. The patient population age ranged from 22–91 years (median 63, IQR 56-71), with 226 males and 223 females. All chest CT scans were obtained using a standard chest CT protocol and were reconstructed using multiple kernels and displayed with a lung window in axial view. A total of 132 patients (29.4%) were diagnosed with UIP, 37 patients (8.2%) with chronic hypersensitivity pneumonitis (CHP), 142 patients

### a) ILD subcategory classification by joint CNN models

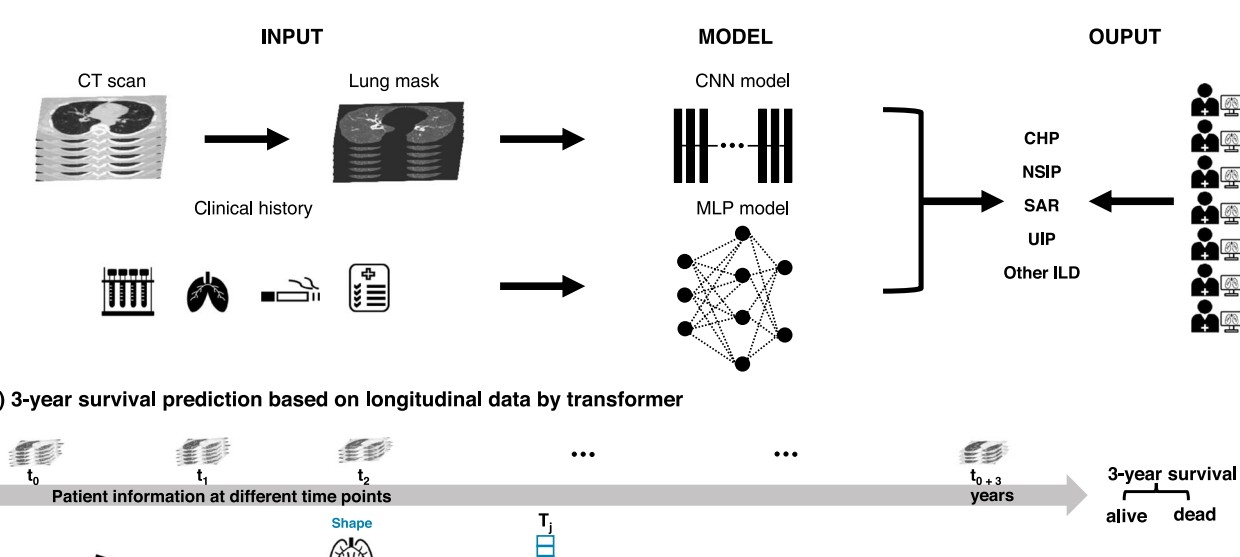

### b) 3-year survival prediction based on longitudinal data by transformer

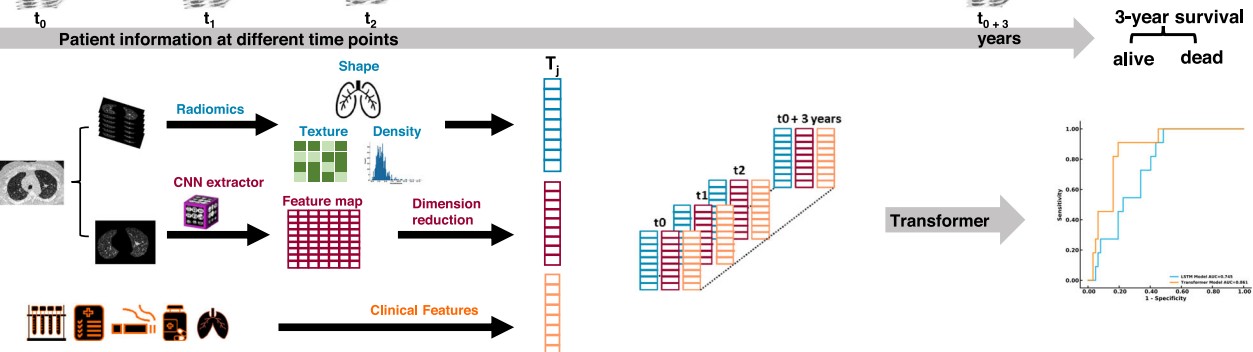

**Fig. 1 | Overview of the framework.** The ILD classification model was generated to predict the subtype of ILD for each patient based on CT scans of the chest and clinical information. A survival rate prediction AI model was generated based on the longitudinal data of each patient. **a** For the classification of ILD, we preprocessed CT scans to obtain the lung regions of each image. Then, we integrated the probability achieved by using a CNN model to study lung images and using an MLP model to study clinical information. Finally, we compared the ILD classification results from the joint AI model with human readers. **b** For the prediction of the 3-year survival rate, the patient information, including image features extracted via Radiomics and CNN model and clinical features, were collected during each visit and then used to generate a Transformer model to predict the risk of each patient.

### a) Patient's characteristics

| | UIP (n=132) | CHP (n=37) | NSIP (n=142) | Sarcoidosis (n=42) | Other ILD (n=96) |
|---|---|---|---|---|---|
| Sex (male) | 91 (68.9%) | 14 (37.8%) | 48 (33.8%) | 28 (66.7%) | 45 (46.9%) |
| [a,b]Age (years) | 68.5±9.3 (62, 76) | 68.5±8.2 (65, 72) | 56.9±12.5 (50, 66) | 53.4±10.8 (45, 61) | 62.0±12.0 (56, 70) |
| Smoking history | | | | | |
| Former smoker | 96 (72.7%) | 17 (45.9%) | 63 (44.4%) | 16 (38.1%) | 55 (57.3%) |
| Current smoker | 4 (3.0%) | 1 (2.7%) | 7 (4.9%) | 0 (0) | 2 (2.1%) |
| Clinical history | | | | | |
| Rheumatic disease | 33 (25.0%) | 3 (8.1%) | 114 (80.3%) | 41 (97.6%) | 43 (44.8%) |
| Home oxygen | | | | | |
| Yes | 70 (53.0%) | 11 (29.7%) | 42 (29.6%) | 6 (14.3%) | 27 (28.1%) |
| Unknown | 2 (1.5%) | 0 (0) | 0 (0) | 0 (0) | 0 (0) |
| Lung biopsy | | | | | |
| Yes | 38 (28.8%) | 12 (32.4%) | 41 (28.9%) | 13 (31.0%) | 40 (41.7%) |
| Unknown | 0 (0) | 1 (2.7%) | 0 (0) | 0 (0) | 0 (0) |
| Occupation exposure | 31 (23.5%) | 10 (27.0%) | 15 (10.6%) | 18 (42.9%) | 24 (25%) |
| Pulmonary hypertension | | | | | |
| Yes | 37 (28.0%) | 16 (43.2%) | 47 (33.1%) | 9 (21.4%) | 19 (19.8%) |
| Unknown | 0 (0) | 1 (2.7%) | 0 (0) | 5 (11.9%) | 3 (3.1%) |
| Lung function test | | | | | |
| [a,b]FEV1 | 1.94±0.63 (1.50, 2.38) | 1.60±0.57 (1.17, 1.98) | 1.86±0.77 (1.37, 2.21) | 2.26±0.91 (1.71, 3.05) | 1.89±0.65 (1.44, 2.32) |
| [a,b]FVC | 2.40±0.83 (1.78, 2.83) | 2.01±0.76 (1.46, 2.54) | 2.35±1.03 (1.71, 2.84) | 3.30±1.07 (2.52, 4.04) | 2.56±0.85 (1.98, 3.00) |
| [a,b]FEV1/FVC | 82.0±8.8 (77, 87) | 80.8±8.9 (77, 87) | 79.8±8.5 (75, 86) | 67.6±15.2 (59, 77) | 74.7±13.7 (69, 83) |
| [a,b]DLCO | 41.3±16.4 (29, 51) | 48.4±17.2 (36, 59) | 52.1±20.9 (34, 68) | 66.4±17.5 (53, 78) | 56.6±21.3 (44, 70) |

### b) Variable correlations to a subcategory

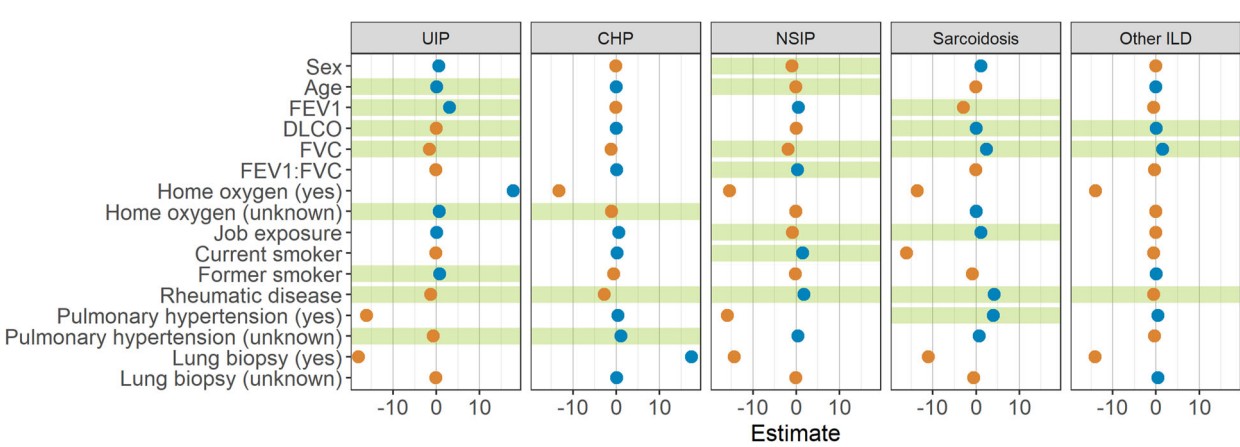

**Fig. 2 | Characteristics and correlations of patient's clinical history.**
**a** Characteristics of patient's clinical information for each ILD subtype. [a]Data in parentheses show interquartile range. [b]Indicates mean ± s.d. Data in parentheses shows the percentage of the population with the characteristic. **b** Correlations between clinical information and each ILD subcategory. The x-axis indicates the coefficient of each clinical variable evaluated by logistic regression. Green shades show a significant correlation.

(31.6%) with NSIP, 42 patients (9.4%) with sarcoidosis and 96 patients (21.4%) with other various ILD. Of the 449 patients in the MSMC-ILD, 128 who had their initial scan and pulmonary function test performed at the Mount Sinai Hospital (MSH) were used as an external testing set. The remaining 321 patients were randomly split into a training set (80.4%, 258 cases with 78 UIP) and a validation set (19.6%, 63 cases with 20 UIP).

We performed a logistic regression with each ILD subcategory as the outcome and the clinical variables as the predictors to determine whether there existed a correlation between the type of ILD and the clinical history. Detailed descriptions and distributions of clinical history are reported in Fig. 2. The logistic regression confirmed that age, FEV1, DLCO, FVC, home oxygen status, former smoking history, history of rheumatic disease, and history of pulmonary hypertension were strongly correlated to UIP ($p = 0.78$). Home oxygen status, history of rheumatic disease, and history of pulmonary hypertension were significant features of CHP ($p = 0.91$). Patient's sex and age, FVC, FEV1/FVC ratio, occupational exposures, former smoking history, and history of rheumatic disease were strongly related to NSIP ($p = 0.096$).

Significant predictors of sarcoidosis were FEV1, DLCO, FVC, occupational exposures, history of rheumatic disease, and history of pulmonary hypertension ($p = 0.99$). Finally, DLCO, FVC, and history of rheumatic disease were key features that correlated with other ILD ($p = 0.31$).

We evaluated the AI models on the unseen external test set. The performance of the joint AI model was compared to seven readers who included a senior thoracic radiologist (STR) with 11 years of experience, two junior thoracic radiologists (JTR1 and JTR2) with 5 years of experience and 4 years of experience, respectively, a thoracic radiology fellow (TRF), two senior general radiologists (SGR1 and SGR2) with specialty in musculoskeletal and 10 years of experience and specialty in pediatric radiology and 15 years of experience respectively, and finally a senior pulmonologist (SP) with 10 years of experience. All readers were provided with the same deidentified lung CT scans and clinical information. The area under the receiver operating characteristic curve (AUROC), sensitivity, and specificity were calculated for each ILD category in our study. The performance and comparison of the AI model and human readers are reported in Fig. 3. Comparisons

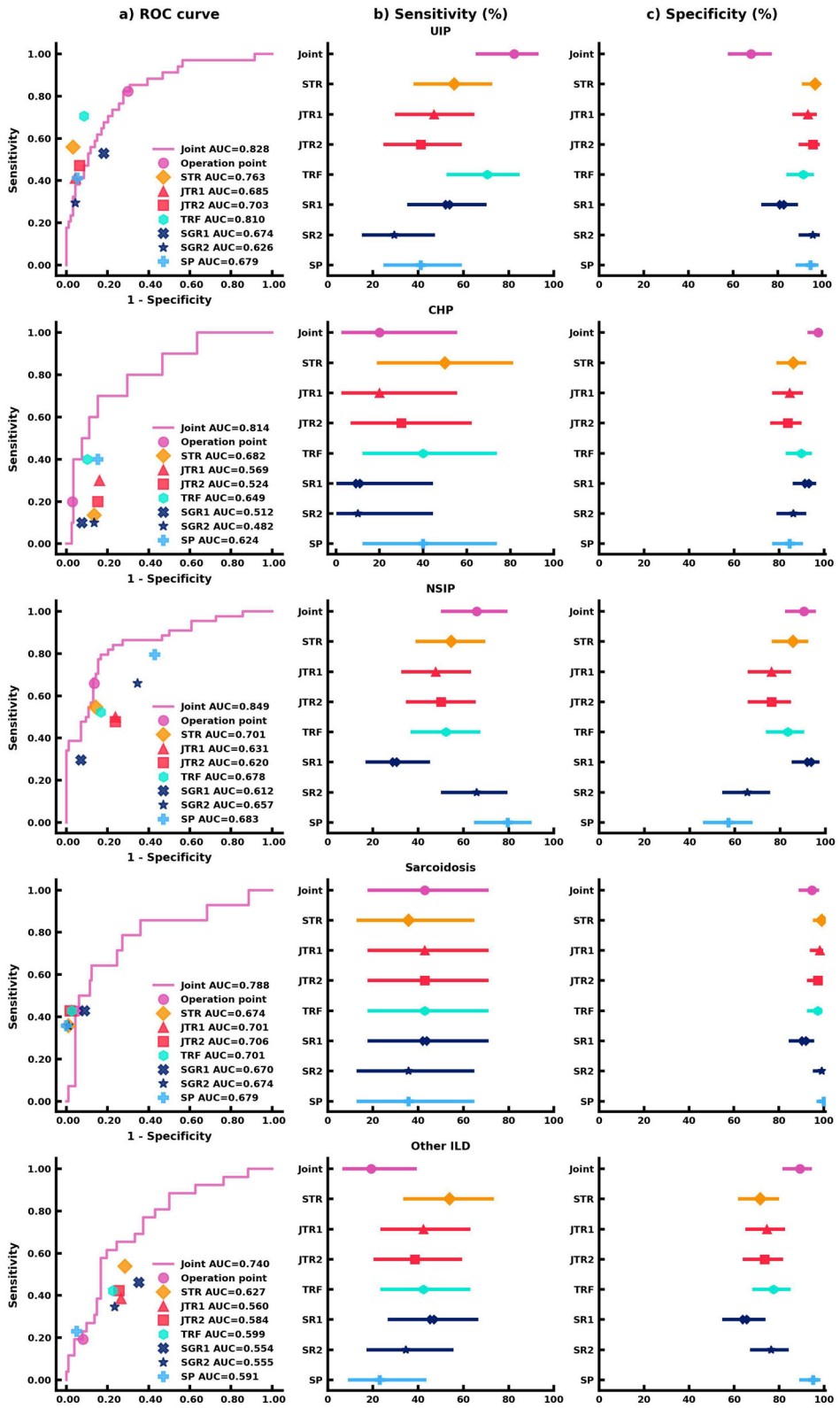

**Fig. 3 | Results of the AI model on the ILD classification. a** AUC comparison between the joint AI model and human readers on the classification of each ILD subtype. The (**b**) Sensitivity analysis of the joint AI model and human readers' results. The markers represent the sensitivity of the AI model and human readers on each ILD subtype, and the lines represent the confidence interval of sensitivity. **c** Specificity analysis of the joint AI model and human readers' results. The markers represent the specificity of the AI model and human readers on each ILD subtype, and the lines represent the confidence interval of specificity. Each human reader was indicated with different markers. Sensitivity and specificity comparison were calculated via the exact Clopper-Pearson method to compute the 95% confidence interval (CI). In (**b**, **c**), data are presented as true sensitivity/specificity +/− 95% CI respectively.

between the joint model, STR, and SP were highlighted hereafter. Detailed performance of the other 5 human readers can be found in Supplementary Tables 1–3.

For UIP classification, the joint model combining CT scans and clinical information had the highest sensitivity in comparison to all human readers, though the human readers had higher specificities. The joint model had a sensitivity of 82.4% (95% confidence interval (CI) 65.5%, 93.2%), an 68.1% specificity (95% CI 57.7%, 77.3%), and an AUROC of 0.828 (95% CI 0.748, 0.909). Importantly, the joint model outperformed the STR (55.9%; $p < 0.05$) and the SP (41.2%; $p < 0.001$) in sensitivity.

For CHP classification, the joint model displayed the highest specificity. The joint model achieved a 20.0% sensitivity (95% CI 2.5%, 55.6%), a 97.5% specificity (95% CI 92.8%, 99.5%), and an AUROC of 0.814 (95% CI 0.676, 0.951), which was equivalent to the STR and SP in sensitivity (50.0%, $p = 0.38$; 40.0%, $p = 0.63$) and significantly better in specificity as compared to both readers (86.4%, $p < 0.01$; 84.8%, $p < 0.001$).

The joint model was more sensitive and specific to classifying and diagnosing NSIP. The joint model achieved a 65.9% sensitivity (95% CI 50.1%, 79.5%), a 90.5% specificity (95% CI 82.1%, 95.8%), and an AUROC of 0.849 (95% CI 0.777, 0.922), which was comparable to the STR who had a 54.6% sensitivity (95% CI 38.9%, 69.6%; $p = 0.33$), an 85.7% specificity (95% CI 76.4%, 92.4%; $p = 0.48$), and an AUROC of 0.701 (95% CI 0.618, 0.785). The SP showed an equivalent 79.6% sensitivity (95% CI 64.7%, 90.2%; $p = 0.11$), but was outperformed in specificity (57.1%; 95% CI 45.9%, 67.9%; $p < 0.001$).

For the classification of sarcoidosis, the joint model and human readers had comparable sensitivities and specificities. The joint model achieved a 42.9% sensitivity (95% CI 17.7%, 71.1%), a 94.7% specificity (95% CI 88.9%, 98.0%), and an AUROC of 0.788 (95% CI 0.643, 0.933). The STR had a 35.7% sensitivity (95% CI 12.8%, 64.9%; $p = 1$), a 99.1% specificity (95% CI 95.2%, 100.0%; $p = 0.13$), and an AUROC of 0.674 (95% CI 0.544, 0.805). The SP had a 35.7% sensitivity (95% CI 12.8%, 64.9%; $p = 1$), a 100.0% specificity (95% CI 96.8%, 100.0%; $p < 0.05$), and an AUROC of 0.679 (95% CI 0.548, 0.809).

While human readers tended to be more sensitive than the joint model in classifying other ILD, the joint model was more specific. The joint model achieved a 19.2% sensitivity (95% CI 6.6%, 39.4%), an 89.2% specificity (95% CI 81.5%, 94.5%), and an AUROC of 0.740 (95% CI 0.636, 0.844). The STR had a 53.9% sensitivity (95% CI 33.4%, 73.4%; $p < 0.05$), a 71.6% specificity (95% CI 61.8%, 80.1%; $p < 0.001$), and an AUROC of 0.627 (95% CI 0.520, 0.734). The SP had a 23.1% sensitivity (95% CI 9.0%, 43.7%; $p = 1$), a 95.1% specificity (95% CI 88.9%, 98.4%; $p = 0.11$), and an AUROC of 0.591 (95% CI 0.506, 0.676).

The Transformer models using longitudinal radiomics and CT scan features and clinical information were used to predict a 3-year survival rate. We extracted 55,296 textual features based on volumetric CT studies. A pretrained CNN model containing underlying CT characteristics was used as an extractor to filter each CT image, and a total of 32 high-level CT features from each study were included. Medication history and other therapeutic information were added to clinical history, bringing the total to 18 clinical variables. A total of 165 features incorporating both imaging and clinical features were assessed longitudinally to create dynamic predictive models in a 3-year survival rate. Detailed descriptions of these 165 features and its correlations with the survival rate were reported in Supplementary Table 4 and details of medications and therapeutic classes were summarized in Supplementary Table 5. Patients having 3-year follow-up information and known living status were included in the progression management study, resulting in 234 participants. Out of 599 visits from 234 patients in the time series analysis, 179 visits from 103 patients contain missing PFTs. We use the nearest visit PFT of each patient as the missing visit PFT. A total of 79 patients only had one visit in our system. The median number of visits within 3 years was 4, and the median time interval between each visit was 8 months. At the end of each year, the estimated mortality rate substantially increased from 2.1% then 6.4% then 9.4%.

We developed models at four endpoints starting from the initial visit to evaluate the patient's response after treatment. Four Transformer models were developed using the patient's initial visit information, and the data within 1 year, 2 years, and 3 years. False negatives were minimized. Only negative predictive value and sensitivity are reported hereafter. More details are demonstrated in Fig. 4. The Transformer models tended to be more predictive with more follow-up data available, showing an uptrend AUROCs of 0.660 (95% CI 44.09%, 87.87%; $p = 0.07981$), 0.632 (95% CI 41.33%, 85.06%, $p = 0.04951$), 0.801 (95% CI 68.92%, 91.19%; $p = 0.153$), and 0.868 (95% CI 77.04%, 96.57%) evaluated at the initial visit, within 1 year, 2 years, and 3 years respectively. All models remained high, with negative predictive values ranging from 89.66 to 94.55%. The models became more sensitive when more follow up information was available, increasing in sensitivity from 54.55 (95% CI 23.38%, 83.25%) to 72.73% (95% CI 39.03%, 93.98%) at the end of year 1 and the end of year 3 respectively.

## Discussion

Diagnosing, treating, and managing interstitial lung disease and its subtypes remains a complex clinical challenge, often requiring the expertize of highly specialized physicians, such as thoracic fellowship-trained radiologists, and the synthesis of an array of clinical information. Lack of human resources and limited access to clinicians with specialized expertise in ILD is a worldwide barrier in ILD management[15]. Furthermore, quantifying a patient's response to treatment and disease progression is a second barrier to clinical care[16]. Walsh[17] et al. developed a deep learning model of 1157 high resolution CT scans to classify UIP and non-UIP. It achieved accuracy 79% in classifying 29 UIP cases. Choe[18] et al. created a content-based image retrieval method to classify four subtypes of ILD, UIP, NSIP, COP and CHP based on CT scans of 288 patients and showed that their proposed framework can improve radiologists' ILD classification accuracy from 52.4% to 72.8%. Both studies only used CT images for algorithm development, while in our study we combine CT images with clinical information together to develop a joint model in order to develop a more comprehensive algorithm to study ILD subtype classification. In addition, we conducted a 3 year survival analysis using longitudinal data of patients to monitor patient's disease progression. Our present study created a joint CNN model by integrating CT images with clinical information. This model accurately predicted five ILD subtypes and outperformed a senior thoracic radiologist and a senior pulmonologist in diagnosing true cases of UIP ($p < 0.05$; $p < 0.001$). Our joint CNN model also performed as well as all human readers in sensitivity when diagnosing CHP, sarcoidosis, NSIP, and other ILD ($p > 0.05$). In addition to the diagnostic joint CNN model, we created a Transformer model that can predict a patient's 3-year survival rate after a visit with high sensitivity and negative predictive value while remaining a reasonably high specificity and positive predictive value.

The joint CNN model showed superior performance in the classification of ILD subtypes. Pretrained weights from the RadImageNet models[19] were used as starting points for CNN. The RadImageNet pretrained model contained CT features such as pulmonary infiltrates. These features shared high-level similarity to our target ILD data, which further improved the CNN performance on CT images. While ViT showed great potential on large natural image datasets[9], the ViT model was outperformed by the CNN model using transfer learning due to the small sample size of images from our ILD dataset (Supplementary Fig. 4). After synthesizing CT images and clinical history and using weights pre-trained from similar studies, the joint CNN was more sensitive to diagnosing UIP that outperformed the STR ($p < 0.05$), JTRs ($p < 0.001$), SGRs ($p < 0.001$), and SP ($p < 0.001$).

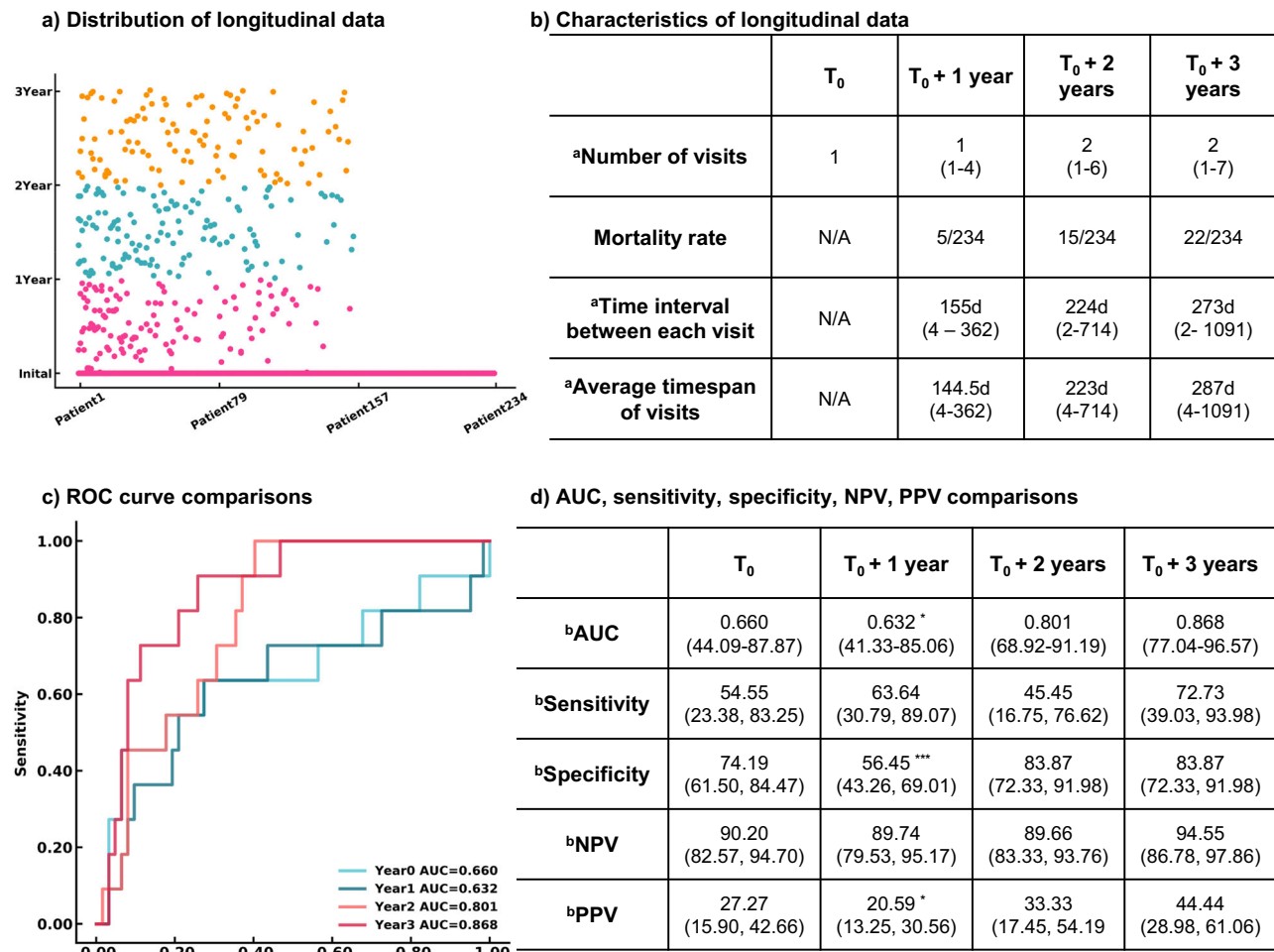

**Fig. 4 | Results of the AI models to predict the 3-year survival rate. a** Distribution of longitudinal visits from each patient. Each visit of 234 patients included in the survival rate study was presented. **b** Characteristics of longitudinal data. **c** ROC curves of 3-year survival rate prediction at different endpoints. **d** Performance and comparison of Transformer models developed at multiple endpoints. $n = 234$ for 3-year survival analysis. Two-sided $P$-values were calculated for all comparisons. [a]Data in parentheses indicate the range. [b]Data in parentheses indicate 95% CI. ***$p < 0.001$. **$p < 0.01$. *$p < 0.05$.

For the diagnosis of CHP, the joint model achieved equivalent performance in sensitivity ($p > 0.05$) to all human readers and outperformed the STR ($p < 0.01$), JTRs ($p < 0.001$), TRF ($p < 0.05$), SGR2 ($p < 0.01$), and SP ($p < 0.001$) in specificity. Regarding NSIP prediction, the joint model performed equally well in sensitivity as compared to six readers ($p > 0.05$) and outperformed SGR1 ($p < 0.01$). Similarly, it demonstrated higher specificity compared to the JTRs ($p < 0.001$), SP ($p < 0.001$), and SGR2 ($p < 0.001$) and performed comparably to the STR ($p = 0.48$). Both the joint model and human readers performed equally well in reading sarcoidosis ($p > 0.05$) in sensitivity and specificity, except that the joint model was outperformed by the SP ($p < 0.05$) in specificity. For the diagnosis of other ILD, the joint model demonstrated similar performance to six human readers ($p > 0.05$) and was only outperformed by the STR ($p < 0.05$); the model was significantly more specific than six human readers ($p < 0.01$) except for the SP ($p = 0.11$).

To analyze the 3-year survival rate, we developed two-time series models, the LSTM and Transformer; both models consisted of multiple factors including quantitative CT information, clinical history, and medication history within 3 years. The average of the Transformer models achieved 7.5% better performance than the average of LSTM models, and the ensemble Transformer model achieved 15.8% better performance than the ensemble LSTM model. Thus, the Transformer algorithm was applied to train patients' data within 1 year, 2 years, and

3 years. The confidence of 3-year survival prediction via Transformer was increased with more follow-up information. The AUROC was dramatically improved by 31.5% between the evaluation at the initial visit and the end of year 3. There was no difference between the evaluation at the end of year 2 and year 3 ($p = 0.153$). This shows that response to treatment may require more than one year. After 2 years of treatment, there is high confidence (95%) in predicting the patient's survival.

Our study sought to address two major barriers in interstitial lung disease management. Firstly, the diagnosis of ILD subtypes often requires thoracic fellowship-trained radiologists, and specialists with such expertize are scarce. This potentially limits the timely diagnosis and treatment of persons living with ILD. Thus, with widespread implementation of our deep learning system, we hope to alleviate the burden on these highly specialized clinicians while enhancing patient care. The implementation of our deep learning system could provide a useful diagnostic tool for the general radiologists in the community who infrequently encounter interstitial lung disease. Instead of simply reporting these diverse disease processes using broad terms such as "pulmonary fibrosis", the general radiologist could use this deep learning system to reach a specific diagnosis. Our diagnostic joint model, for example, showed superior sensitivity in identifying UIP ($p < 0.05$) and significant improvement in specificity for the diagnosis of CHP ($p < 0.05$), NSIP ($p < 0.05$), and other ILD ($p < 0.05$) as

compared to human readers. The second barrier in interstitial lung disease management is disease prognosis and progression. It is important to evaluate treatment efficacy and patient prognosis at each visit so that patients may be counseled about their condition and what to expect. Accurately predicting patient response and prognosis is extremely challenging but has great value by ultimately improving patient outcomes. Our Transformer model can evaluate the 3-year survival rate at each visit by integrating information from each visit. Moreover, the Transformer model demonstrated significant advancement in predicting a 3-year survival rate when current follow-up information was integrated. Our deep learning system has the potential to be integrated into the daily workflow of pulmonologists, rheumatologists, pathologists, and radiologists, where it could serve as a second opinion for a diagnosis of ILD subtypes and dynamically provide personalized insights regarding current and future treatment efficacy using its 3-year survival prediction feature. Installation of the deep learning models would require cloud computing with the integration of PACS and Epic or other clinical databases, which is relatively easy to achieve in most modern healthcare systems.

Our proposed deep learning system has limitations. One major limitation is that a patient's initial visit in the registry may not be the patient's first evaluation for ILD since our patients come from multiple areas. For the unknown values in the categorical variables, we made an additional class within each variable to indicate them. In the ILD classification part, our lung segmentation algorithm uses a high threshold, so some opacity in the image might be missed. In the time series study, one limitation is a small sample size since 79 of 234 patients had only one visit. Because we split the training, validation, and test dataset based on different hospital resources, the prevalence of death differs between the test dataset and training/validation datasets which have around 10% death in the whole sample, making it more difficult for the model to study the characteristics from the deceased patients. In addition, deaths were determined from the chart declaration. The causes of death might not be only associated with ILD. Lastly, during preprocessing clinical variables for the time series study, missing PFTs were filled with the same data from the nearest visit.

In future studies, more clinical history and additional clinical data, including symptoms after treatment and long-term survival rate, can be analyzed when further follow-up information is gathered. We also aim to collect pathology slides and genetic data to comprehensively diagnose ILD subtypes and improve treatment and outcomes. Deploying the models in a cloud setting could help clinicians access the results faster. The reproducibility of the models needs further evaluation at multiple medical centers.

In conclusion, the proposed deep learning system demonstrates high potential in accurately diagnosing five subtypes of ILD. This could help clinicians without access to specialized thoracic training fellow, to diagnose and make dynamic predictions regarding patient prognosis and disease progression. We believe the proposed models, which integrate CT images with clinical history, demonstrate equivalent performance to a senior thoracic radiologist and a senior pulmonologist and also evaluate survival rate at each follow-up visit, which could be a useful tool to distinguish ILD subcategories and manage the long-term progression of patients.

## Methods

### Ethics oversight

The study was approved by the Institutional Review Board (IRB) of the Mount Sinai School of Medicine, in accordance with Mount Sinai's Federal Wide Assurances to the Department of Health and Human Services (ID# STUDY-14-00584-CR001). Written informed consent has been obtained from patients enrolled in this research registry. A Data and Safety Monitoring Board (DSMB) from Mount Sinai IRB had oversight of the study.

### Study population

We collected chest CT scans and clinical information from 458 patients enrolled in the MSMC-ILD between September 2014 and April 2021. Individuals for participation in Mount Sinai Medical Center Research Registry for Interstitial Lung Disease (MSMC-ILD) included all adult (age > 18 years old) patients who were receiving or seeking medical care for the treatment of interstitial lung disease at Mount Sinai Medical Center, St Luke's and Beth Israel Medical Centers. Patients with lung fibrosis or other interstitial lung disease were enrolled in the MSMC-ILD and assessing the extent of the disease. MSMC-ILD was established in 2014. The diagnosis of an ILD subtype followed the ATS2018 guidelines. All registry patients had a consensus diagnosis from radiology, pathology, and pulmonology. In this study, occupational exposure or other environmental exposure is included as a clinical feature. It is likely that the patient cohort at MSMC might be different from other patient cohorts. For example, patients at MSMC might be influenced by World Trade Center exposure. There were nine patients excluded due to low image quality resulting in a total of 449 patients with both clinical information and CT images that were included in our ILD diagnosis study. The patient population age ranged from 22 to 91 years (median 63, IQR 56-71), with 226 males and 223 females. A total of 132 patients (29.4%) were diagnosed with usual interstitial pneumonia (UIP), 37 patients (8.2%) with chronic hypersensitivity pneumonitis (CHP), 142 patients (31.6%) with nonspecific interstitial pneumonia (NSIP), 42 patients (9.4%) with sarcoidosis and 96 patients (21.4%) with other various ILD. 234 patients were selected for the 3-year survival analysis (see Supplementary Fig. 1 for inclusion and exclusion criteria). Sex information was used in the diagnosis of ILD subtypes as well as the prediction of 3 year survival analysis. Study participants did not receive compensation.

### Clinical information

Clinical information was retrospectively collected by medical students, radiology residents, and thoracic radiology fellows through chart review via electronic medical records. The following data were collected within 6 months of the study date of each patient's CT scan: age, sex, history of current or former smoking, history of rheumatic disease, home oxygen requirement, history of occupational or other exposures (including pets), World Trade Center exposure, pulmonary function test (PFT) values (FEV1/FVC ratio, FEV1 value, DLCO percentage), presence of pulmonary hypertension based on echocardiography or right heart catheterization, and history/results of lung biopsy. Clinical information was collected from pulmonology visit notes in the Electronic Medical Record and PFT flowcharts. If data was not available within the 6-month time frame, the data entry for that variable was left blank. Incomplete clinical variables were later filled with values from the nearest visit.

We also recorded the medications being used at or about the time of the CT to treat the ILD. There were eight types of medicine used for patients, including azathioprine (immunosuppressant), bosentan (cardiovascular), cyclophosphamide (antineoplastics), mycophenolate (immunosuppressant), nintedanib (unclassified), pirfenidone (unclassified), prednisone (hormone), rituximab (unclassified).

### Data split

The dataset was split by patient ID and hospital. For ILD subtype classification, 128 (28.5%) patients with their initial CT scan and clinical information collected at the Mount Sinai Hospital were used as the external test set. The rest of the 321 (71.5%) patients whose initial data were collected at outside hospitals were used for model development, with 258 (57.5%) patients within the training set and 63 (14.0%) patients into the validation set. For the analysis of the 3-year survival rate, a subset of 234 patients meeting the criteria in Supplementary Fig. 1 was utilized. These 234 patients were split into a training dataset

containing 123 patients (6 dead), a validation dataset containing 38 patients (5 dead), and a test dataset containing 73 patients (11 dead).

## Human reader studies

The predictions of the joint CNN AI model were compared to seven human readers on the test set. Six board-certified and fellowship-trained radiologists and a pulmonologist, as well as one thoracic radiology fellow, were provided with the same initial CT scan and associated clinical information that were used to develop the AI models. A senior thoracic radiologist (A.J.) with 10 years of post graduate experience, two junior thoracic radiologists (M.C. and A.B.) with 5-years post graduate experience, a thoracic radiology fellow (A.S.), two senior radiologists (M.H. and J.M.) with 10 years of experience in non thoracic radiology specialties(musculoskeletal radiology and pediatric radiology respectively), and a senior pulmonologist (S.D.) with 10 years experience each reviewed the 128 studies and associated clinical information from the test set. Their predictions were compared to the predictions of the joint deep learning model and the consensus diagnosis.

## AI models

The consensus diagnosis of UIP, CHP, NSIP, sarcoidosis, and other various ILD was used as the gold standard to develop the AI models in ILD subcategory classification. We created five models using image data and clinical information. First, a CNN model (model 1) using pre-trained weights from the RadImageNet[19] and ViT model (model 2) based on CT images were developed. Second, machine learning models (model 3), including MLP, SVM, and XGBoost, were generated based on the clinical information. Finally, a joint CNN model (model 4) and a joint ViT model (model 5) were developed which integrated both the imaging and clinical data.

**ILD subtype classification model training.** We used the same optimization strategies for all classification AI models by employing the Adam optimizer with a learning rate of 0.001 and weight decay of 0.0001, except model 4 used a learning rate of 0.0001. Each model was trained with 40 epochs. We used categorical cross-entropy as the objective function.

To predict patients' 3-year survival rate longitudinal radiological and clinical data were used to create time series models based on each time point (initial visit, year 1, year 2, year 3) with LSTM[14] and Transformer[13], respectively.

**Three-year survival rate model training.** We used the same optimization strategies for all longitudinal AI models by employing the Adam optimizer of learning rate of 0.0001 and weight decay of 0.0001. Both LSTM and the Transformer were developed in two different parameter settings. For each parameter setting, we repeated the simulation 30 times. Each simulation was trained with 100 epochs with a batch size of 64. We used categorical cross-entropy as the objective function. The top two models from each parameter setting with the best performance on the validation dataset were selected for the ensemble model. A total of four models through averaging probability for each patient were then calculated for their performance on the test dataset. The details of parameter settings were reported in later sections.

## Data preprocessing

**Clinical information and CT data collection.** The following clinical data were collected: patients' sex, age, lung function lab test results (FEV1, DLCO, FVC, FEV1/FVC), smoking history, occupational exposure, rheumatic disease, hypertension, lung biopsy, and the use of home oxygen. CT imaging data were collected from the study DICOM header. For missing data, we added an unknown class to each categorical variable. The LabelEncoder function in the scikit-learn package was utilized to encode these categorical data into numerical variables.

The StandardScaler function in the scikit-learn package was used to normalize each feature to unit variance with the mean set as 0.

**Image preprocessing.** First, all CT scans were resampled to an isotropic voxel. Next, we generated lung regions for each image in each study. This was achieved by applying a threshold of -400HU to each CT slice to effectively convert the CT image into a binary image consisting of two densities—air and not air. The "not air" periphery of the binary image was removed, and the two largest "air" regions were kept. The binary mask was then used on the input raw CT image to separate the lung regions. After lung segmentation, a standard lung window (width = 1500HU and level = -400HU) was used to normalize pixel intensities between 0 and 255 for each segmented lung CT slice. GE Centricity Universal Viewer 6.0 was used to review the CT studies. Preprocessed images were used to develop CT-based models in Tensorflow (2.4.0).

## CT-based convolutional neural network model (model 1)

We designed a CT-based convolutional neural network model to diagnose ILD using the CT images. This CT-based CNN model was built via transfer learning using pre-trained weights from a RadImageNet convolutional neural network Inception-ResNet-V2 (IRV2)[19–21]. We froze all layers from the pre-trained model and only trained the top10 layers that incorporated high-level features. An average pooling layer and the last dense classifier layers were followed by the last convolutional layer. Using a RadImageNet pre-trained model provided a better starting point than an ImageNet model as the RadImageNet database contains CT lung images and therefore shares higher similarity with the target data.

## CT-based vision Transformer model (model 2)

We trained a CT-based vision Transformer model. ViT model was developed to transfer the success of the self-attention mechanism on NLP tasks into imaging applications[9]. Our ViT model first split the input image into 10 patches and encoded each embedded patch into a self-attention based deep neural network. Then, two fully connected layers with 2048 and 1024 nodes and the final prediction layer were followed by the encoded embedding layers.

## Machine learning model (model 3)

To classify ILD subtypes based on clinical information, we applied MLP, SVM, and XGBoost classifiers to build machine learning models. We evaluated the performance of these three classifiers on the validation dataset (Supplementary Fig 5). We fine-tuned the model's hyperparameters on the training and validation dataset and evaluated the best model on the test dataset. For the SVM classifier, we assessed the 'C' and kernel type. For the XGBoost classifier, the learning rate and several iterations were tuned. For MLP, we assessed the number of hidden nodes in each layer, the learning rate, activation method, and solver for weight optimization. After the hyperparameter optimization, the two-layer MLP model with 64 and 32 nodes was selected because it achieved the highest AUC score on the validation dataset.

## Joint CNN and MLP model (model 4)

A joint model combining CT images and clinical information was developed. The inception-res-net-v2 architecture using pre-trained weights derived from the RadImageNet database[19] was used to learn features from imaging data. Given the pre-trained weights included CT imaging patterns relevant to our targeted CT images we froze the base layers that stored fundamental information from CT features and only trained the top10 layers that incorporated high-level features. An average pooling layer and three full layers with 1024, 512, and 32 nodes were followed by the last convolutional layer. CT images were finally presented in a vector with 32 features. 16 clinical variables were learned by the MLP model that had two fully connected layers with 64

and 32 nodes, respectively. The last MLP layer was combined with the vector containing CT features. The joint vector was then fed into a fully connected layer having 512-dimensional features before the output layer.

### Joint ViT and MLP model (model 5)

A joint ViT and MLP model was also developed to study the combined information of CT images and clinical data. Because the location of lung regions might vary in CT images from different centers, we chose to split the input image into 32 patches. Then, patches were processed via the Transformer encoder, which contained four independent self-attention heads to repeat the computation in parallel. The image features extracted from the Transformer encoder were then connected with three fully connected layers with 1024, 512, and 32 nodes. All CT images were presented in a vector with 32 features. Similar to model 4, a total of 16 clinical variables were learned by the MLP model that had two fully connected layers with 64 and 32 nodes, respectively. The last layer of the MLP model was combined with the vector containing CT features. The joint vector was then fed into a fully connected layer having 512-dimensional features before the output layer.

### Radiomics

Radiomics was used to extract textual features of normal lung regions from CT images[22]. We first converted our segmented lung CT images into binary images as the masked images to indicate the region of interest for Radiomics. Then, we applied the PyRadiomics tool to combine CT images and masked CT images to obtain textual features based on volumetric data. The features extracted from PyRadiomics contain information about the size, shape, spatial relationship, and image intensity of medical images[23]. A total of 116 radiomics features were obtained for further model development in predicting a 3-year survival rate.

### CNN extractor and Uniform Manifold Approximation and Projection (UMAP)

We used a pre-trained RIN-generic IRV2 CNN model developed on the RadImageNet database as the extractor to obtain high-level CT features. The last convolutional layer conv_7b having 1536 kernel maps in 6 by 6 matrix size, was used to screen each CT image. Each CT image was presented as a vector of 55,296 features. We then averaged the CT slices from each study. After CNN feature extraction, we used UMAP[24] to reduce the dimension of features while preserving the global structure allowing the 55, 296 features to be reduced to 32.

### Time Series data preprocessing

The time-series data included clinical information, medication information, and imaging features for each patient visit were extracted from Radiomics and CNN. The MinMaxScaler function was used to normalize all features. The maximum visit number from our dataset was seven, so patients who had less than seven visits were given data values of zero for the "missing" visits as the sign for our model to skip the data during processing. Sklearn (0.24.1) was used to preprocess and develop the models.

### Transformer time series model

We developed a Transformer time series model to study the temporal information from the time series data of patients' clinical information and CT image features. The Transformer time series model was developed by stacking 16 Transformer encoders together to evaluate data at each time point. The time-series data were processed via the Transformer encoders and then followed by an average pooling layer and a fully connected layer with 128 nodes. We fine-tuned the hyperparameters of the Transformer

model on the training and validation dataset and evaluated the best model on the test dataset. We assessed the number of heads in the Transformer encoder.

### LSTM time series model

LSTM is an improved form of a Recurrent Neural Network, designed to solve the problem of vanishing long-term gradients[14]. The LSTM time series model was developed to predict living status based on patients' clinical information and CT image features over time. The time-series input was first passed through two layers of LSTM, which computes the corresponding sequence of input data at different time states and then outputs a sequence of hidden state vectors in forward and reverse directions. Then, the features extracted from LSTM were followed by three fully connected layers and one final classifier layer. We fine-tuned the hyperparameters of the LSTM model on the training and validation dataset and evaluated the best model on the test dataset. We assessed the number of LSTM layers.

### Statistical analysis

Comparisons of AUROCs were performed by bootstrap in the pROC package (version 1.18.0)[25] in R. A total of 2000 bootstrap permutations were simulated to calculate 95% CIs and p-values. The 95%CIs of sensitivity and specificity for AI models and human readers were calculated by the exact Clopper-Pearson method[26]. McNemar's test[27] was used to compare sensitivity and specificity. Generalized score statistic test[28] was used to calculate p values for negative predictive values and positive predictive values. Two-sided p values were assessed for all statistical analyses, and p-value < 0.05 was defined as statistical significance. We performed logistic regression to evaluate the correlations between clinical variables and each ILD subcategory. The Hosmer–Lemeshow test[29] confirmed the goodness of logistic regression. McNemar's and the generalized score statistic tests were performed in the DTComPair[30] package (version 1.0.3) in R 4.1.3.

### Reporting summary

Further information on research design is available in the Nature Portfolio Reporting Summary linked to this article.

## Data availability

The in-house datasets generated and/or analysed during the current study are not publicly available due to HIPAA compliance and were used with Mount Sinai institutional permission for the purposes of this project. All requests for access to in-house data will be addressed to the corresponding authors, Dr. Xueyan Mei (xueyan.mei@icahn.mssm.edu), Dr. Yang Yang (yang.yang4@ucsf.edu) or Dr. Zahi Fayad (zahi.fayad@mssm.edu), and will be processed in accordance with Mount Sinai institutional guidelines. Mount Sinai Innovation Partners (MSIP) will assess all requests based on the purpose of data request, and it may take up to one month to process the request. A material-transfer or data-usage agreement will be required between Mount Sinai and the receiving organization. The requesting organization must provide comprehensive details, including the name and full contact information of the individual and institution making the request, along with specific identification of the data being requested. Additionally, the requesting organization must clearly state the intended purpose of the data transfer and provide assurances that the transferred data will only be used for non-commercial academic and educational purposes in compliance with Mount Sinai institutional guidelines. The pretrained models used in this paper are available at https://doi.org/10.1148/ryai.210315. Source data are provided as a Source Data file. Source data are provided with this paper.

## Code availability

All the codes we used to train the models have been posted in this github repository https://github.com/lzl199704/ILD.

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

## Acknowledgements

X.M. was supported by the National Center for Advancing Translational Sciences (NCATS) TL1TR004420 NRSA TL1 Training Core in Transdisciplinary Clinical and Translational Science (CTSA).

## Author contributions

X.M. and Z.L. developed the modeling. X.M., Z.L. and J.Q.X.G. created the figures. X.M., Z.L., A.S., M.L., P.B., C.D., C.C, T.D., M.P., A.J., Z.A.F. and Y.Y. wrote the manuscript. X.M., M.P., M.C., A.B., A.J., Z.A.F. and Y.Y. designed the experiments. A.S., M.C., A.B., A.J., S.D., J.M., and M.H. evaluated and read the test set cases. M.L, A.S., P.B., J.L., C.D., S.P., G.S., and B.G. collected the dataset. X.M. and Z.L. performed the statistical analysis. X.M, Z.A.F and Y.Y. supervised the work.

## Competing interests

T.D. is managing partner of RadImageNet LLC.
