## [Peer review file · Nature Communications]

REVIEWER COMMENTS

Reviewer #1 (Remarks to the Author):

This is an interesting paper addressing a topic of utmost relevance. In the last few years there has been an increasing number of publications dealing with AI and diagnosis of ILD. Several papers have already showed how AI software can outperform even experienced radiologists in diagnosing ILD. These software are also showing their potential in the prognostic assessment of patients with ILD but so far there have been very few if any attempt to develop AI software integrating imaging and clinical data to assess prognosis of ILD patients.

The results of this paper are exciting, nevertheless there are some points that should be improved.

In the paper the Authors classify the ILD in UIP, NSIP, CHP, Sarcoidosis and other. In daily practice ILD patients are usually classified according to the disease rather than to the pathological or radiological pattern. UIP is usually the radiological and pathological pattern of Idiopathic pulmonary fibrosis: do the Authors consider UIP as synonymous of IPF? The same is true for NSIP: do the authors consider NSIP as synonymous of CTD related ILD?

This represents a limit for the paper that should be clarified.

The Authors state that patients enrolled in MSMC-ILD had a consensus diagnosis from radiology, pathology, and pulmonology. It would be interesting to know in how many of these patients a lung biopsy was performed and to know how the software performed in this group of patients.

The Authors report the prognostic value of their software in terms of 3 years survival, nevertheless they report that also an evaluation of patient's response after treatment was performed. This aspect should be clarified: was the prognosis assessed only in terms of overall survival or also in terms of treatment response (e.g. assessment of FVC decline in patients undergoing antifibrotic treatment)?

In the discussion the Authors report that quantitative CT information have been considered to analyze the 3 year survival rate, nevertheless there is no mention of any quantitative CT analysis in the paper. This aspect should be clarified.

The results of the software are interesting but it should be better explained which clinical data showed higher value in the prognostic assessment of these patients.

Reviewer #2 (Remarks to the Author):

Mei et al aimed to develop and validate several deep learning tools using real-world data from patients with ILD to 1.) classify ILD subtypes based on baseline CT scans and clinical history and 2.) predict three-

year mortality using CT and clinical information in response to treatment. A CNN model using CT scans and clinical factors was able to accurately subcategorize amongst UIP, CHP, NSIP, sarcoidosis, and other ILD, with best performance in diagnosing UIP, outperforming several senior thoracic radiologists and a senior pulmonologist. The most compelling aspect of this work for me is the prediction of three-year survival using the Transformer model, as patients can often have large variations in morbidity and mortality even with the same underlying ILD diagnosis, and clinicians often cannot accurately prognosticate clinical course nor treatment success. Thus, an accurate deep learning algorithm using imaging and clinical data has the possibility of helping physicians provide more accurate diagnoses and prognoses to their patients. The training and testing cohorts used in this manuscript came from real-world patients seen at an ILD center, highlighting the potential use of deep learning algorithms directly within hospital EMR and PACS systems, although the single center raises concerns for generalizability of the results. This interesting study was well thought out, and the manuscript was clear with easy-to-follow writing. I have some comments to help the reader more fully understand the study design and results.

1. The cohort data came from the time period between September 2014 and April 2021. How did the authors account for any confounders of the COVID-19 pandemic that may bias their results, especially when predicting 3-year survival, since we know that patients with underlying lung disease are at higher risk of COVID-19 and of worse subsequent sequelae including death? Additionally, how did the authors account for any COVID-19 infections amongst participants in their cohort that would also increase parenchymal changes like pulmonary infiltrates on a CT scan and further bias the results?
2. It would be helpful to know how many in the cohort had missing data in the time series study, especially for important data points like PFTs since they are commonly used by clinicians for ILD disease monitoring and prognostication.
3. The time series models were used to evaluate the patients' response after treatment. It would be helpful to see the breakdown of treatments used for each ILD, especially since the choice of the second-line or even first-line treatment can vary for ILDs and impact subsequent morbidity and mortality.
4. How were the deaths adjudicated? What were the causes of death?
5. What was the inter-reader variability amongst the readers for each of the ILD subtypes? The consensus (or lack thereof) of the different readers is important to consider, since the gold standard for ILD diagnosis is not with a CT chest read by a single radiologist or pulmonologist but rather with a multidisciplinary discussion of radiologists, pathologists, pulmonologists, and/or rheumatologists.
6. What was the rationale for using -400 HU for segmentation? As the authors mentioned in the Discussion, some opacity is likely to be missed. I am considering the studies of high attenuation areas (HAA) between -600 HU and -250 HU, which have been shown to be associated with interstitial abnormalities, worse lung function, and mortality, and to be a risk factor for ILD (PMID: 28613921). While I recognize that HAA has been studied in participants without underlying ILD, you may be nonetheless be losing some meaningful CT features from your ILD patients by setting your binary cutoff at -400 HU.

7. What was the rationale for including radiomics features in addition to the CT scan features when doing the time series analysis? Why not also include them in the joint CNN and ViT models for ILD subtyping?

Reviewer #3 (Remarks to the Author):

Overall the paper is well written. Longitudinal analysis of clinical data and CT images is clinically important to dynamically predict lung disease progression and prognosis and predict the survival rate. Collecting large data for this study is challenging due to the nature of longitudinal data. In this study, the authors collected a large number of longitudinal data for analyzing survival rates with AI models. Also, the authors show the AI models for the diagnosis of interstitial lung disease (ILD). Comprehensive experiments have been conducted to analyze the effectiveness of AI models. I think the paper has high merit in the field of clinical analysis of interstitial lung diseases with AI. The following are the summaries of strengths and weaknesses to further improve the manuscript.

Strengths

- Detailed information regarding the data collection is provided. Patient inclusion and exclusion criteria are provided.
- Various AI models are compared for two tasks (ILD classification and survival rate prediction)
- Statistical analysis is conducted for experiments
- The performance of AI models is compared with an expert reader study. This can provide a really important discussion.

Weaknesses

- The distribution of training(+validation)/test set is missing. It would be great to provide the distribution of training and test datasets for both tasks (ILD classification and survival rate prediction)
- Image quality (resolution) needs to be improved.
- It seems that Human Reader Study was conducted by the authors. It might induce optimistic biases. It will be great to clarify the procedures for human reader study.
- Lots of different models are used in this study but the detail of some models are missing. It might be challenging to reproduce the study due to the limited information. The following are examples: How to train ViT model? Did the authors use a pre-trained model as in CNN? What were the selected hyperparameters for SVM? How did the authors design the Joint CNN and MLP model (There are lots of

design choices)? Why feature concatenation is used for combining two different features? What is the detailed architectures for the Transformer time series model and LSTM time series model, respectively?

- Qualitative examples are missing. It will be great to provide examples images (successful cases and failure cases)

- The dataset and code are not provided.

- Three-year survival rate model training: For the following sentence, “Both LSTM and the Transformer were developed in two different parameter settings”, which parameter settings are analyzed for each model?

- Comparison with other studies is missing. It will be great to add discussion with other studies.

- Extended Data Fig. 6: Why step size of Figure c is largely different from Figure a and Figure b?

REVIEWER COMMENTS

Reviewer #1 (Remarks to the Author):

This is an interesting paper addressing a topic of outmost relevance. In the last few years there has been an increasing number of publications dealing with AI and diagnosis of ILD. Several paper have already showed how AI software can outperform even experienced radiologists in diagnosing ILD. These software are also showing their potential in the prognostic assesment of patients with ILD but so far there have been very few if any attempt do develop AI software integrating imaging and clinical data do assess prognosis of ILD patients.

The results of this paper are exciting, nevertheless there are some points that should be iproved.

1. In the paper the Authors classify the ILD in UIP, NSIP, CHP, Sarcoidosis and other. In daily practice ILD patients are usually classified according to the disease rather than to the pathological or radiological pattern. UIP is usually the radiological and pathological pattern of Idiopathic pulmonary fibrosis: do the Authors consider UIP as synonymous of IPF? The same is true for NSIP: do the authors consider NSIP as synonymous of CTD related ILD? This represent a limit for the paper that should be clarified.

We thank the reviewer for the comments. In this paper, UIP is not considered as synonymous with IPF, nor is NSIP considered as synonymous with CTD. Consensus of histopathologic and radiologic results were used as the standard for ILD subtyping in this study. Patients enrolled in the MSMC-ILD registry had a consensus diagnosis from radiology, pathology, and pulmonology. We used the pathologic and radiologic patterns as the classification basis instead of the disease classification system. In addition, as UIP is not always idiopathic and NSIP is not always related to CTD, we did not use these terms interchangeably.

2. The Authors state that patients enrolled in MSMC-ILD had a consensus diagnosis from radiology, pathology, and pulmonology. It would be interesting to know in how many of these patients a lung biopsy was performed and to know how the software performed in this group of patients.

We thank the reviewer for the comment. We summarized the distribution of clinical data in Figure 2 which showed the percentage of lung biopsies performed. We break down the performance of this group in both ILD subtype diagnosis and survival analysis in Table R1 and Table R2.

Table R1. Deep learning performance regarding lung biopsy status in ILD subtype diagnosis

	UIP	CHP	NSIP	Sar	Other ILD
--	-----	-----	------	-----	-----------

Lung Biopsy (Yes, n= 36)	0.775	0.618	0.808	0.909	0.677
Lung Biopsy (No, n= 92)	0.851	0.874	0.859	0.754	0.779
P value	0.407	0.304	0.592	0.118	0.380

Patients who underwent lung biopsy showed higher AUC in 3-year survival analysis. P value was calculated via the pROC package in R and bootstrap replicate number was set to 1000. This suggests that lung biopsy is not a significant predictor in ILD subtype diagnosis.

Table R2. Deep learning performance regarding lung biopsy status in 3-year survival analysis

	AUC
Lung Biopsy (Yes, n= 21)	1.0
Lung Biopsy (No, n= 52)	0.810
P value	<0.05 (0.0037)

The deep learning model performed significantly better for patients undergoing lung biopsy ($p < 0.05$). The lung biopsy might provide additional information in prognosis assessment that could potentially result in a better prediction of the 3-year survival rate.

- The Authors report the prognostic value of their software in terms of 3 years survival, nevertheless they report that also an evaluation of patient's response after treatment was performed. This aspect should be clarified: was the prognosis assessed only in terms of overall survival or also in terms of treatment response (e.g. assessment of FVC decline in patients undergoing antifibrotic treatment)?

We thank the reviewer for the comment. The prognosis was assessed in terms of overall survival and treatment response. We have added experiments using treatment features (the medication label and therapeutic label). The transformer model with treatment inputs achieved an AUC of 0.933 and the transformer model without treatment inputs achieved an AUC of 0.891 ($p = 0.603$) (Figure R1). This suggests that treatment of the radiomics features, clinical history, and patients demographics play an important role in prognosis assessment and treatment features can make contributions to improve the performance but are not significant. We hope this is sufficient for the reviewer.

Figure R1. Comparison of models with and without treatment.

4. In the discussion the Authors report that quantitative CT information have been considered to analyze the 3 year survival rate, nevertheless there is no mention of any quantitative CT analysis in the paper. THIS aspect should be clarified. The results of the software are interesting but it should be better explained which clinical data showed higher value in the prognostic assessment of these patients.

We thank the reviewer for this comment. There are a total of 165 features used for the development of the 3-year survival analysis model. We performed a logistic regression with all features that we used to assess prognosis to determine whether there existed a correlation between the quantitative features and survival rate. There are five features (diagnostics_Image_original_Minimum, original_shape_VoxelVolume, original_firstorder_Range, original_firstorder_Maximum, original_firstorder_TotalEnergy) having perfect separation situation, and It turned out that the following features are statistically significant in analyzing the 3 year survival rate with p value less than 0.05 (Hosmer-Lemeshow Goodness of Fit $p = 0.435$), including FormerSmoker, age, average timespan, therapeutic label, original_firstorder_InterquartileRange, original_firstorder_Variance, original_glcm_Imc2, original_gldm_GrayLevelVariance, original_glrIm_GrayLevelNonUniformity, original_glrIm_GrayLevelNonUniformityNormalized, original_glrIm_RunPercentage, original_glszm_GrayLevelNonUniformity, CNN_feature_2, and CNN_feature_19. Detailed descriptions of the quantitative features and its coefficient scores are reported in Extended Data Table 4 as follows.

features	p value	features	p value	features	p value	features	p value
Home_O2	0.182	original_firstorder_Median	0.579	original_gldm_LargeDependenceLowGrayLevelEmphasis	0.688	original_ngtdm_Contrast	0.825
Occupation	0.652	original_firstorder_Minimum	0.315	original_gldm_LowGrayLevelEmphasis	0.292	original_ngtdm_Strength	0.102
CurrentSmoker	0.276	original_firstorder_RobustMeanAbsoluteDeviation	0.816	original_gldm_SmallDependenceEmphasis	0.381	age	0.042
FormerSmoker	0.033	original_firstorder_RootMeanSquared	0.492	original_gldm_SmallDependenceHighGrayLevelEmphasis	0.750	kvp	0.165
Hx_disease	0.703	original_firstorder_Skewness	0.967	original_gldm_SmallDependenceLowGrayLevelEmphasis	0.253	manufacturer	0.614
pulm_HTN	0.785	original_firstorder_Uniformity	0.139	original_glrlm_GrayLevelNonUniformity	0.020	thickness	0.311
Biopsy	0.564	original_firstorder_Variance	0.014	original_glrlm_GrayLevelNonUniformityNormalized	0.008	CNN_feature_0	0.768
sex	0.864	original_glcm_Autocorrelation	0.217	original_glrlm_GrayLevelVariance	0.475	CNN_feature_1	0.847
Med_label	0.144	original_glcm_ClusterProminence	0.317	original_glrlm_HighGrayLevelRunEmphasis	0.524	CNN_feature_2	0.024
Therapeutic_label	0.007	original_glcm_ClusterShade	0.098	original_glrlm_LongRunEmphasis	0.965	CNN_feature_3	0.633
FEV1	0.852	original_glcm_ClusterTendency	0.160	original_glrlm_LongRunHighGrayLevelEmp	0.953	CNN_feature_4	0.052

				hasis			
FVC	0.899	original_glcm_Contrast	0.160	original_glrlm_LongRunLowGrayLevelEmphasis	0.947	CNN_feature_5	0.099
FEV1.FVC	0.187	original_glcm_Correlation	0.829	original_glrlm_LowGrayLevelRunEmphasis	0.068	CNN_feature_6	0.320
DLCO	0.972	original_glcm_DifferenceAverage	0.211	original_glrlm_RunEntropy	0.523	CNN_feature_7	0.246
timespan	0.798	original_glcm_DifferenceEntropy	0.276	original_glrlm_RunLengthNonUniformity	0.147	CNN_feature_8	0.484
diagnostics_Image.original_Mean	0.185	original_glcm_DifferenceVariance	0.189	original_glrlm_RunLengthNonUniformityNormalized	0.980	CNN_feature_9	0.231
diagnostics_Image.original_Maximum	0.743	original_glcm_Id	0.234	original_glrlm_RunPercentage	0.033	CNN_feature_10	0.177
diagnostics_Mask.original_VoxelNumber	0.729	original_glcm_Idm	0.237	original_glrlm_RunVariance	0.950	CNN_feature_11	0.075
diagnostics_Mask.original_VolumeNumber	0.471	original_glcm_Idmn	0.419	original_glrlm_ShortRunEmphasis	0.117	CNN_feature_12	0.543
original_shape_Elongation	0.761	original_glcm_Idn	0.590	original_glrlm_ShortRunHighGrayLevelEmphasis	0.422	CNN_feature_13	0.873
original_shape_Flatness	0.566	original_glcm_Imc1	0.743	original_glrlm_ShortRunLowGrayLevelEmphasis	0.180	CNN_feature_14	0.320
original_shape_LeastAxis	0.748	original_glcm_Imc2	0.042	original_glszm_GrayLevelNo	0.035	CNN_feature_15	0.061

sLength				nUniformity			
original_shape_MajorAxisLength	0.706	original_glcm_InverseVariance	0.233	original_glszm_GrayLevelNonUniformityNormalized	0.707	CNN_feature_16	0.572
original_shape_Maximum2DDiameterColumn	0.725	original_glcm_JointAverage	0.177	original_glszm_GrayLevelVariance	0.156	CNN_feature_17	0.603
original_shape_Maximum2DDiameterRow	0.725	original_glcm_JointEnergy	0.393	original_glszm_HighGrayLevelZoneEmphasis	0.744	CNN_feature_18	1.000
original_shape_Maximum2DDiameterSlice	0.917	original_glcm_JointEntropy	0.687	original_glszm_LargeAreaEmphasis	0.675	CNN_feature_19	0.029
original_shape_Maximum3DDiameter	0.743	original_glcm_MCC	0.679	original_glszm_LargeAreaHighGrayLevelEmphasis	0.415	CNN_feature_20	0.148
original_shape_MeshVolume	0.748	original_glcm_MaximumProbability	0.293	original_glszm_LargeAreaLowGrayLevelEmphasis	0.402	CNN_feature_21	0.612
original_shape_MinorAxisLength	0.792	original_glcm_SumAverage	0.177	original_glszm_LowGrayLevelZoneEmphasis	0.428	CNN_feature_22	0.249
original_shape_Sphericity	0.558	original_glcm_SumEntropy	0.060	original_glszm_SizeZoneNonUniformity	0.050	CNN_feature_23	0.226
original_shape_SurfaceArea	0.980	original_glcm_SumSquares	0.160	original_glszm_SizeZoneNonUniformityNormalized	0.547	CNN_feature_24	0.544
original_shape_SurfaceVolumeRatio	0.088	original_gldm_DependenceEntropy	0.909	original_glszm_SmallAreaEmphasis	0.565	CNN_feature_25	0.164

original_first_order_10Percentile	0.292	original_gldm_DependenceNonUniformity	0.648	original_glszm_SmallAreaHighGrayLevelEmphasis	0.788	CNN_feature_26	0.914
original_first_order_90Percentile	0.240	original_gldm_DependenceNonUniformityNormalized	0.199	original_glszm_SmallAreaLowGrayLevelEmphasis	0.547	CNN_feature_27	0.297
original_first_order_Energy	0.075	original_gldm_DependenceVariance	0.756	original_glszm_ZoneEntropy	0.624	CNN_feature_28	0.734
original_first_order_Entropy	0.282	original_gldm_GrayLevelNonUniformity	0.239	original_glszm_ZonePercentage	0.933	CNN_feature_29	0.322
original_first_order_InterquartileRange	0.018	original_gldm_GrayLevelVariance	0.020	original_glszm_ZoneVariance	0.681	CNN_feature_30	0.291
original_first_order_Kurtosis	0.865	original_gldm_HighGrayLevelEmphasis	0.223	original_ngtdm_Busyness	0.354	CNN_feature_31	0.210
original_first_order_MeanAbsoluteDeviation	0.633	original_gldm_LargeDependenceEmphasis	0.058	original_ngtdm_Coarseness	0.165	time_filter	0.580
original_first_order_Mean	0.531	original_gldm_LargeDependenceHighGrayLevelEmphasis	0.892	original_ngtdm_Complexity	0.943	average_timespan	0.034

Extended Data Table 4: Detailed descriptions of the quantitative features and its correlation with 3-year survival rate.

Reviewer #2 (Remarks to the Author):

Mei et al aimed to develop and validate several deep learning tools using real-world data from patients with ILD to 1.) classify ILD subtypes based on baseline CT scans and clinical history and 2.) predict three-year mortality using CT and clinical information in response to treatment. A CNN model using CT scans and clinical factors was able to accurately subcategorize amongst UIP, CHP, NSIP, sarcoidosis, and other ILD, with best performance

in diagnosing UIP, outperforming several senior thoracic radiologists and a senior pulmonologist. The most compelling aspect of this work for me is the prediction of three-year survival using the Transformer model, as patients can often have large variations in morbidity and mortality even with the same underlying ILD diagnosis, and clinicians often cannot accurately prognosticate clinical course nor treatment success. Thus, an accurate deep learning algorithm using imaging and clinical data has the possibility of helping physicians provide more accurate diagnoses and prognoses to their patients. The training and testing cohorts used in this manuscript came from real-world patients seen at an ILD center, highlighting the potential use of deep learning algorithms directly within hospital EMR and PACS systems, although the single center raises concerns for generalizability of the results. This interesting study was well thought out, and the manuscript was clear with easy-to-follow writing. I have some comments to help the reader more fully understand the study design and results.

1. The cohort data came from the time period between September 2014 and April 2021. How did the authors account for any confounders of the COVID-19 pandemic that may bias their results, especially when predicting 3-year survival, since we know that patients with underlying lung disease are at higher risk of COVID-19 and of worse subsequent sequelae including death? Additionally, how did the authors account for any COVID-19 infections amongst participants in their cohort that would also increase parenchymal changes like pulmonary infiltrates on a CT scan and further bias the results?

We thank the reviewer for this comment. We did not account for any confounder of COVID-19 patients who developed fibrosis features. For patients who enrolled in the registry, ILD fibrosis was the primary diagnosis. We confirmed that patients who had follow-ups between March 2020 and April 2021 did not have COVID-19.

2. It would be helpful to know how many in the cohort had missing data in the time series study, especially for important data points like PFTs since they are commonly used by clinicians for ILD disease monitoring and prognostication.

We thank the reviewer for this comment. We added this information to the manuscript as follows:

“Out of 599 visits from 234 patients in the time series analysis, 179 visits from 103 patients contain missing PFTs. We use the nearest visit PFT of each patient as the missing visit PFT”.

3. The time series models were used to evaluate the patients’ response after treatment. It would be helpful to see the breakdown of treatments used for each ILD, especially since the choice of the second-line or even first-line treatment can vary for ILDs and impact subsequent morbidity and mortality.

We thank the reviewer for the comment. The prognosis was assessed in terms of overall survival and treatment response. We have added experiments using treatment features

(the medication label and therapeutic label). The transformer model with treatment inputs achieved an AUC of 0.933 and the transformer model without treatment inputs achieved an AUC of 0.891 ($p = 0.603$) (Figure R1). This suggests that treatment of the radiomics features, clinical history, and patients demographics play an important role in prognosis assessment. We hope this is sufficient for the reviewer.

Figure R1. Comparison of models with and without treatment.

4. How were the deaths adjudicated? What were the causes of death?

We determined death from the chart declaration. We did not record the exact cause of death from death certificates given that patients who have died with ILD could be listed with multiple morbidities and all could be correlated to death. For example, an ILD patient with hypertension could expire from urosepsis, and it would not be uncommon for the death certificate to indicate the primary cause of death was urosepsis and secondary hypertension. ILD might not be even mentioned. Therefore, we looked for the patient's deceased information in the chart review. In the limitation, we added that "*Deaths were determined from the chart declaration. The causes of death might not be only associated with ILD.*" We hope this is sufficient for the reviewer.

5. What was the inter-reader variability amongst the readers for each of the ILD subtypes? The consensus (or lack thereof) of the different readers is important to consider, since the gold standard for ILD diagnosis is not with a CT chest read by a single radiologist or pulmonologist but rather with a multidisciplinary discussion of radiologists, pathologists, pulmonologists, and/or rheumatologists.

We thank the reviewer for the suggestion. We calculated the inter-reader variability. Readers with thoracic radiology training showed the most interreader agreement when classifying interstitial lung diseases based on CT images and clinical history.

Figure R2. Interreader Agreement using Cohen's κ based on classifying ILD using a combination of both clinical history and CT images.

6. What was the rationale for using -400 HU for segmentation? As the authors mentioned in the Discussion, some opacity is likely to be missed. I am considering the studies of high attenuation areas (HAA) between -600 HU and -250 HU, which have been shown to be associated with interstitial abnormalities, worse lung function, and mortality, and to be a risk factor for ILD (PMID: 28613921). While I recognize that HAA has been studied in participants without underlying ILD, you may be nonetheless be losing some meaningful CT features from your ILD patients by setting your binary cutoff at -400 HU.

We thank the reviewer for this comment. We applied two thresholds -600HU and -250 HU for lung region segmentation, and then compared the results of models developed based on two thresholds with models based on -400HU lung segmentation images. The following two figures (Figure R3 and Figure R4) indicate that choosing a threshold between -600HU and -250HU has no significant influence on the CT CNN model and joint CNN model. We hope this is sufficient for the reviewer.

Figure R3. CNN model comparisons in different window settings.

Figure R4. Joint model comparison in different window settings.

7. What was the rationale for including radiomics features in addition to the CT scan features when doing the time series analysis? Why not also include them in the joint CNN and ViT models for ILD subtyping?

We thank the reviewer for this comment. The radiomics features and CT scan features are complementary as imaging features in the prediction of survival rate. The radiomics features contain both low-level and high-level in shape, volume, pixel intensity and spatial information. The CT scan features were extracted from a pretrained CNN model. The last convolutional layer of the CNN model contained more than 50,000 high-level features (localization and spatial) but was flattened to a vector in order to feed into the output layer, thereby some spatial information was lost. To take advantage of this, we combined both radiomics features and high-level CT scan features along with clinical history to assess prognosis.

For ILD subtype diagnosis, the input image is equivalent to the input information of the radiomics features and CT scan features. In addition, the convolutional neural networks learned both temporal and spatial features from the bottom (lower level local features such as color and shape) to the top layers (higher level global features such as spatial information). Thus, we did not use radiomics features in the ILD subtype diagnosis.

Reviewer #3 (Remarks to the Author):

Overall the paper is well written. Longitudinal analysis of clinical data and CT images is clinically important to dynamically predict lung disease progression and prognosis and predict the survival rate. Collecting large data for this study is challenging due to the nature of longitudinal data. In this study, the authors collected a large number of longitudinal data for analyzing survival rates with AI models. Also, the authors show the AI models for the diagnosis of interstitial lung disease (ILD). Comprehensive experiments have been conducted to analyze the effectiveness of AI models. I think the paper has high merit in the field of clinical analysis of interstitial lung diseases with AI. The following are the summaries of strengths and weaknesses to further improve the manuscript.

Strengths

- Detailed information regarding the data collection is provided. Patient inclusion and exclusion criteria are provided.
- Various AI models are compared for two tasks (ILD classification and survival rate prediction)
- Statistical analysis is conducted for experiments
- The performance of AI models is compared with an expert reader study. This can provide a really important discussion.

Weaknesses

1. The distribution of training(+validation)/test set is missing. It would be great to provide the distribution of training and test datasets for both tasks (ILD classification and survival rate prediction)

We thank the reviewer for this comment. We reported the distribution of training/validation/test sets for both tasks in Extended Data Fig. 1.

Extended Data Fig. 1 | Patient inclusion and exclusion criteria.

2. Image quality (resolution) needs to be improved.

We thank the reviewer for this comment. We updated the dpi of our figures. It might be compressed in the initial submission. All the figures we submitted have 300 dpi with higher quality.

3. It seems that Human Reader Study was conducted by the authors. It might induce optimistic biases. It will be great to clarify the procedures for human reader study.

We thank the reviewer for this comment. We first de-identified the chest CT scans and clinical information. For example, MRNs and accession numbers were replaced with dummy variables. Therefore, all readers were blind to the true diagnosis. The readers were also provided with de-identified clinical information. We mimicked the clinical workflow when the human readers reviewed the studies, that is, the presence of chest CT images and clinical history at the same time. The deep learning model achieved

similar performance to individual human readers, maybe because the deep learning model was trained with the consensus label, which was confirmed by radiologist, pulmonologist, and pathologist. Meanwhile, all readers participating in this study were asked to review the studies independently. We hope this is sufficient for the reviewer.

4. Lots of different models are used in this study but the detail of some models are missing. It might be challenging to reproduce the study due to the limited information. The following are examples: How to train ViT model? Did the authors use a pre-trained model as in CNN? What were the selected hyperparameters for SVM? How did the authors design the Joint CNN and MLP model (There are lots of design choices)? Why feature concatenation is used for combining two different features? What is the detailed architectures for the Transformer time series model and LSTM time series model, respectively?

We thank the reviewer for this comment. The ViT model was trained from scratch based on our ILD CT images, and we didn't use any pretrained weights as there were no available RadImageNet pretrained weights in transformers. The architecture of the ViT model was mentioned as *"Our ViT model first split the input image into 10 patches and encoded each embedded patch into a self-attention based deep neural network. Then, two fully connected layers with 2048 and 1024 nodes and the final prediction layer were followed by the encoded embedding layers"*. To train this ViT model, we evaluate the hyperparameters of the number of heads in the attention layer and the learning rate of the model.

The hyperparameters for SVM model in 3-year survival analysis include the regularization parameter C of 1, 10, 100, or 1000, and the choice of kernel type, including linear, poly, and rbf.

The idea of the joint CNN and MLP model is to combine the knowledge from CT images learned by CNN model and clinical information learned by MLP model. Thus, to train a network that can learn two parallel inputs, image and clinical information, we concatenate the last layer of CNN model with the last layer of MLP model which both contain 32 nodes. During the training of this joint model, this concatenate layer serves as the layer to adjust weights for preferring image knowledge or clinical knowledge while classifying different ILD subtypes.

The detailed architecture of the Transformer time series model is as follows: the main model contains the 16 stack of transformer encoder blocks which consists of layer normalization, multi-head attention and dropout, the global average pooling layer to reduce the output tensor, and the final fully connected layer with 128 nodes. The hyperparameters for the transformer time series model are the choice of the number of heads in the transformer encoder block. Using multiple heads in the transformer can help us form a more consistent 3-year survival analysis.

The detailed architecture of the LSTM time series model is as follows: The time-series input was first passed through two layers of LSTM, which computes the corresponding sequence

of input data at different time states and then outputs a sequence of hidden state vectors in forward and reverse directions. Then, the features extracted from LSTM were then followed by three fully connected layers and one final classifier layer. The LSTM model was fine-tuned by the number of nodes in the LSTM layer.

5. Qualitative examples are missing. It will be great to provide examples images (successful cases and failure cases)

We thank the reviewer for this comment. We added the sample cases as follows to demonstrate the performance of readers and AI models.

Figure R5. A) Axial noncontrast CT of the chest in a 53 year old male demonstrates peripheral reticular densities with associated traction bronchiectasis and focal honeycombing at the left lung base. This case was diagnosed as UIP by the senior chest radiologist, one junior chest radiologist and the chest fellow. It was diagnosed as UIP by the joint model. The consensus diagnosis was UIP. B) Axial noncontrast CT of the chest in a 76 year old female demonstrates mild poorly defined ground glass densities along the bronchovascular bundles in the mid lungs with mild distortion of the parenchyma and mild bronchiectasis. This case was diagnosed as NSIP by two junior chest radiologists and the senior radiologist. It was diagnosed as Other ILD by the joint model. The consensus diagnosis was NSIP. C) Axial noncontrast CT of the chest in a 73 year old female demonstrates peripheral lower lobe ground glass opacities with associated architectural distortion and bronchiectasis. A patulous esophagus is also present. This case

was diagnosed as Other ILD by one senior chest radiologist, two junior chest radiologists, one chest fellow, and two senior radiologists. It was diagnosed as NSIP by the joint model. The consensus diagnosis was NSIP. D) Axial noncontrast CT of the chest in a 63-year-old female demonstrates mild multi-station non-calcified lymphadenopathy. There were no significant pulmonary findings. This case was diagnosed as NSIP by one junior chest radiologist, one senior radiologist and one senior pulmonologist, and as Other ILD by one senior chest radiologist, one junior chest radiologist, one chest fellow and one senior radiologist. It was diagnosed as NSIP by the joint model. The consensus diagnosis was sarcoidosis.

6. The dataset and code are not provided.

We thank the reviewer for this comment. All the codes we used to train the models have been posted in this github repository <https://github.com/lzl199704/ILD>, and the data will be available by reasonable request to the corresponding authors.

7. Three-year survival rate model training: For the following sentence, “Both LSTM and the Transformer were developed in two different parameter settings”, which parameter settings are analyzed for each model?

We thank the reviewer for this comment. As the detail architecture described in #4, the two parameter settings for the Transformer models are the choice of number of head 4 or 6, and the two parameter settings for the LSTM models are 2048 and 1024 or 2048 and 512 nodes in two LSTM layers respectively.

8. Comparison with other studies is missing. It will be great to add discussion with other studies.

We thank the reviewer for this comment. We added more comparison with other studies in the discussion. As shown in the original manuscript, “Prior studies have implemented deep learning methods to diagnose UIP, and proper UIP[17] as well as four ILD subtypes based on chest CT scans, highlighting a potential role for AI in the field.” We added “*Walsh, et al. [17] developed a deep learning model of 1157 high resolution CT scans to classify UIP and non-UIP. It achieved an accuracy of 79% in classifying 29 UIP cases. Choe et al. [18] created a content-based image retrieval method to classify four subtypes of ILD, UIP, NSIP, COP and CHP based on CT scans of 288 patients and showed that their proposed framework can improve radiologists’ ILD classification accuracy from 52.4% to 72.8%. Both studies only used CT images for algorithm development, while in our study we combine CT images with clinical information together to develop a joint model in order to develop a more comprehensive algorithm to study ILD subtype classification. In addition, we conducted a three year survival analysis using longitudinal data of patients to monitor patient’s disease progression.” We hope this is sufficient for the reviewer.”*

9. Extended Data Fig. 6: Why step size of Figure c is largely different from Figure a and Figure b?

We thank the reviewer for this comment. Figure c was the ensemble from the best two models from LSTM and Transformer respectively, while Figure a and b showed the average of 30 simulated scenarios on LSTM and Transformer. An average of 30 models would generate more cutoff options that could result in different sensitivity and specificity when constructing the receiver operating characteristics curve. We hope this is sufficient for the reviewer.

REVIEWER COMMENTS

Reviewer #2 (Remarks to the Author):

Thank you for your responses.

Minor comments:

1.) Please provide, either within Figure 2a or as a supplement, the breakdown of the treatments that were used to treat the patients within each ILD subtype (ie. what proportion of the patients with UIP received pirfenidone vs nintedanib, with NSIP received prednisone, azathioprine, mycophenolate, etc). It would be helpful to know the breakdown since medication and therapeutic labels were additional factors included in the survival models.

2.) Figure 2a line 16 should read "Pulmonary hypertension."

Reviewer #3 (Remarks to the Author):

The revision has well addressed my concerns.

Reviewer 1

"Regarding comment and response 3: This response is not entirely clear. Please clarify whether treatment features were included in the joint models. The authors state that prognosis included treatment response, and I see med_label and therapeutic_label in Extended Data Table 4. However, the authors' comparisons in response 3 and Figure R1, which state, "We have added experiments using treatment features," and show no significant differences in the AUCs, seem to suggest that the treatment features were not actually included in the prognostic models.

- If treatment features were not included in the final models, the fact that treatment features do not make significant contributions to performance of predicting survival should be included as a point of discussion and a limitation, as there is evidence for mortality benefit associated with specific treatments (e.g. antifibrotics for UIP). Furthermore, these lines should be revised in the Discussion: "This shows that response to treatment may require more than one year. After two years of treatment, there is high confidence (95%) in predicting the patient's survival," and "provide personalized insights regarding current and future treatment efficacy," as they are overstated.

- If treatment features were in fact included in the final models, please state that more clearly in the text.

- Lastly, please clarify whether all 165 features were included in Extended Data Table 4, or if only some of them were used."

We thank the reviewer for the comments. In our previous response to the question 3, we meant we have performed additional experiments WITHOUT treatment in 3-year survival analysis for comparison of model with treatment. The model with treatment inputs achieved an AUC of 0.933 and the transformer model without treatment inputs achieved an AUC of 0.891. This suggests that treatment of the radiomics features, clinical history, and patients demographics play an important role in prognosis assessment and treatment features can make contributions to improve the performance. The prognosis model included treatment and therapeutic information in the analysis of the manuscript. All 165 features included in Extended Data Table 4 were included in the prognosis model development.

Reviewer 2

1) Please provide, either within Figure 2a or as a supplement, the breakdown of the treatments that were used to treat the patients within each ILD subtype (ie. what proportion of the patients with UIP received pirfenidone vs nintedanib, with NSIP received prednisone, azathioprine, mycophenolate, etc). It would be helpful to know the breakdown since medication and therapeutic labels were additional factors included in the survival models.

We thank the reviewer for the comment. We have added the breakdown of medications and therapeutic classes within each ILD subtype as Extended Data Table 5. In the manuscript, we modified the text to "*Detailed descriptions of these 165 features and its correlations with the survival rate were reported in **Extended Data Table 4** and details of medications and therapeutic classes were summarized in **Extended Data Table 5.***"

Extended Data Table 5 | Details of medications and therapeutic classes within each ILD subtype.

Medications (Therapeutic Class)	UIP (n=132)	CHP (n=37)	NSIP (n=142)	SAR (n=42)	Other (n=96)
Azathioprine (immunosuppressant)					
Yes	1 (0.1%)	1 (2.7%)	5 (3.5%)	0 (0%)	0 (0%)
No	131 (99.2%)	36 (97.3%)	137 (96.5%)	42 (100%)	96 (100%)
Bosentan (Cardiovascular)					
Yes	0 (0%)	0 (0%)	1 (0.7%)	0 (0%)	0 (0%)
No	132 (100%)	37 (100%)	141 (99.3%)	42 (100%)	96 (100%)
Cyclophosphamide (Antineoplastics)					
Yes	0 (0%)	0 (0%)	1 (0.7%)	0 (0%)	0 (0%)
No	132 (100%)	37 (100%)	141 (99.3%)	42 (100%)	96 (100%)
Mycophenolate (immunosuppressant)					
Yes	2 (1.5%)	4 (10.8%)	25 (17.6%)	0 (0%)	8 (8.3%)
No	130 (98.5%)	33 (89.2%)	117 (82.4%)	42 (100%)	88 (91.7%)
Nintedanib (Other)					
Yes	10 (7.6%)	0 (0%)	0 (0%)	0 (0%)	0 (0%)
No	122 (92.4%)	37 (100%)	142 (100%)	42 (100%)	96 (100%)
Pirfenidone (Other)					
Yes	15 (11.4%)	1 (2.7%)	0 (0%)	0 (0%)	1 (1.0%)
No	117 (88.6%)	36 (97.3%)	142 (100%)	42 (100%)	95 (99.0%)
Prednisone (Hormone)					
Yes	17 (12.9%)	10 (27.0%)	25 (17.6%)	8 (19.0%)	16 (16.7%)
No	115 (87.1%)	27 (73.0%)	117 (82.4%)	34 (81.0%)	80 (83.3%)
Rituximab (Other)					
Yes	0 (0%)	0 (0%)	1 (0.7%)	0 (0%)	0 (0%)
No	132 (100%)	37 (100%)	141 (99.3%)	42 (100%)	96 (100%)
Other (Other)					
Yes	87 (65.9%)	21 (56.8%)	84 (59.2%)	34 (81.0%)	71 (74.0%)
No	45 (34.1%)	16 (43.2%)	58 (40.8%)	8 (19.0%)	25 (26.0%)

2) Figure 2a line 16 should read “Pulmonary hypertension.”

We thank the reviewer for the suggestion. We have modified the entry of Figure 2a line 16 to “pulmonary hypertension”.

a) Patient's characteristics

	UIP (n=132)	CHP (n=37)	NSIP (n=142)	Sarcoidosis (n=42)	Other ILD (n=96)
Sex (male)	91 (68.9%)	14 (37.8%)	48 (33.8%)	28 (66.7%)	45 (46.9%)
^{a,b} Age (years)	68.5±9.3 (62, 76)	68.5±8.2 (65, 72)	56.9±12.5 (50, 66)	53.4±10.8 (45, 61)	62.0±12.0 (56, 70)
Smoking history					
Former smoker	96 (72.7%)	17 (45.9%)	63 (44.4%)	16 (38.1%)	55 (57.3%)
Current smoker	4 (3.0%)	1 (2.7%)	7 (4.9%)	0 (0)	2 (2.1%)
Clinical history					
Rheumatic disease	33 (25.0%)	3 (8.1%)	114 (80.3%)	41 (97.6%)	43 (44.8%)
Home oxygen					
Yes	70 (53.0%)	11 (29.7%)	42 (29.6%)	6 (14.3%)	27 (28.1%)
Unknown	2 (1.5%)	0 (0)	0 (0)	0 (0)	0 (0)
Lung biopsy					
Yes	38 (28.8%)	12 (32.4%)	41 (28.9%)	13 (31.0%)	40 (41.7%)
Unknown	0 (0)	1 (2.7%)	0 (0)	0 (0)	0 (0)
Occupation exposure					
Pulmonary hypertension	31 (23.5%)	10 (27.0%)	15 (10.6%)	18 (42.9%)	24 (25%)
Pulmonary hypertension					
Yes	37 (28.0%)	16 (43.2%)	47 (33.1%)	9 (21.4%)	19 (19.8%)
Unknown	0 (0)	1 (2.7%)	0 (0)	5 (11.9%)	3 (3.1%)
Lung function test					
^{a,b} FEV1	1.94±0.63 (1.50, 2.38)	1.60±0.57 (1.17, 1.98)	1.86±0.77 (1.37, 2.21)	2.26±0.91 (1.71, 3.05)	1.89±0.65 (1.44, 2.32)
^{a,b} FVC	2.40±0.83 (1.78, 2.83)	2.01±0.76 (1.46, 2.54)	2.35±1.03 (1.71, 2.84)	3.30±1.07 (2.52, 4.04)	2.56±0.85 (1.98, 3.00)
^{a,b} FEV1/FVC	82.0±8.8 (77, 87)	80.8±8.9 (77, 87)	79.8±8.5 (75, 86)	67.6±15.2 (59, 77)	74.7±13.7 (69, 83)
^{a,b} DLCO	41.3±16.4 (29, 51)	48.4±17.2 (36, 59)	52.1±20.9 (34, 68)	66.4±17.5 (53, 78)	56.6±21.3 (44, 70)

b) Variable correlations to a subcategory

Fig. 2 | Characteristics and correlations of patient's clinical history.

REVIEWERS' COMMENTS

Reviewer #2 (Remarks to the Author):

The authors have adequately addressed my comments. Thank you.